# Temporal Logic-Based Multi-Vehicle Backdoor Attacks against Offline RL Agents in End-to-end Autonomous Driving

## Abstract

End-to-end autonomous driving (AD) systems integrate complex decision-making processes. Assessing the safety of these systems against potential security threats, including backdoor attacks, is a stepping stone for real-world deployment. However, traditional methods focus on static triggers, which do not adequately reflect the dynamic nature of these systems and could be impractical to deploy in the real world. To address these limitations, we propose a novel backdoor attack against the end-to-end AD systems that leverage multi-vehicles' trajectories as triggers. We employ different behavior models and their configurations to generate the trigger trajectories, which are then quantitatively evaluated using temporal logic specifications. This evaluation guides the subsequent perturbations to the behavior model configurations. Through an iterative process of regeneration and re-evaluation, we can refine and generate realistic and plausible trigger trajectories that involve multiple vehicles' complex interactions. Furthermore, we develop a negative training strategy by incorporating patch trajectories that share similarities with the triggers but are designated not to activate the backdoor. We thus enhance the stealthiness of the attack, refining the system's responses to trigger scenarios. Through extensive empirical studies using offline reinforcement learning (RL) driving agents with various trigger patterns and target action designs, we demonstrate the flexibility and effectiveness of our proposed attack, showing the under-exploration of existing end-to-end AD systems' vulnerabilities to such multi-vehicle-based backdoor attacks. We also evaluate the attack against existing defenses and validate different design choices of our attack via a comprehensive ablation study.

## 1 Introduction

As end-to-end autonomous driving (AD) (Hu et al., 2023; Shao et al., 2023b;a) systems increasingly demonstrate promising performance in diverse real-world applications (Coelho & Oliveira, 2022; Xu et al., 2024), their deployment demands rigorous testing to ensure reliability against a range of security threats (Wang et al., 2021a; Ding et al., 2023a), including backdoor attacks. Recent works have explored backdoor attacks against different modules within AD systems (Pourkeshavarz et al., 2024; Zhang et al., 2024; Ni et al., 2024), while the vulnerabilities within the end-to-end systems have not been fully explored yet. Moreover, as discussed in Gong et al. (2024), traditional backdoor attacks often employ static patterns as triggers which fail to capture the dynamic complexities of real-world driving environments. Multi-vehicle interactions have rarely been considered as potential triggers for backdoor attacks in end-to-end AD systems.

Recognizing this gap, our motivation is to investigate the vulnerabilities of end-to-end AD systems against backdoor attacks with triggers that could be feasibly deployed in real-world situations. We consider a practical scenario where attackers manipulate the behavior of one or multiple vehicles by driving along specific trajectories. These movements can be detected by the poisoned ego car's sensor and LiDAR, thereby activating the backdoor within its system and causing it to crash or behave unpredictably. Notably, such a trigger would be more practical to deploy, compared to the cases of the patch trigger, which might require the sudden appearance of a physical object on the road. Figure 1 illustrates our proposed attack, where the two attacker vehicles simultaneously bypass the ego car and complete the maneuver. The entire process is observed

by the ego car, which has been compromised by the attacker. As a result, the poisoned ego car takes the target action of suddenly turning left, leading to a potentially dangerous outcome.

In this paper, we propose trajectory as the trigger in backdoor attacks against the end-to-end AD systems. We begin by automatically generating the trigger trajectories, with the help of the flexible behavior models and the temporal logic-based evaluation. This also allows us to seamlessly integrate our poisoning process with negative training, avoiding false activation of the target behavior. To the best of our knowledge,

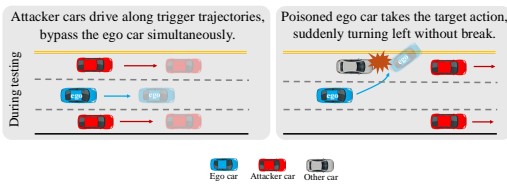

Figure 1: An example of our proposed backdoor attack.

this is the first work demonstrating the practicality of trajectory-based backdoor attacks against end-to-end autonomous driving systems. To demonstrate the feasibility and effectiveness of our approach, we employ offline reinforcement learning-based driving agents. Through extensive experiments, we validate the effectiveness of our proposed attack, showing that they can indeed compromise AV operations under certain conditions. Our contributions can be summarized as follows:

- We introduce a novel approach to backdoor attacks against end-to-end AD systems by utilizing trajectories as the trigger. We design a novel framework that can automatically generate trajectories with complex interactions between multiple attack vehicles, leveraging temporal logic (TL) to evaluate and iteratively refine these trajectories.
- We propose to perform negative training such that the poisoned model's backdoor gets activated only when exact trigger trajectories appear, enhancing the stealthiness of the attack. Our TL evaluation can be seamlessly integrated with the negative training technique, facilitating the generation of a diverse set of trajectory scenarios that are similar yet sufficiently distinct to effectively train the model to recognize only trigger conditions.
- Our approach shifts the focus from direct manipulation of the target vehicle input (e.g., through camera adversarial inputs) to exploiting the vehicle's contextual awareness algorithms. Our attack mirrors potential real-world attack scenarios where an attacker might control one or more vehicles in the vicinity of a target. This highlights an under-explored area in end-to-end autonomous vehicle system vulnerabilities.

## 2 RELATED WORK

**Backdoor attacks in AD systems.** Backdoor attacks have been extensively studied in computer vision (CV) and natural language processing (NLP) domain (Liu et al., 2018; Cheng et al., 2021; Tao et al., 2024; Chou et al., 2023; Li et al., 2021), for AD systems, these attacks specifically target modules that utilize deep learning algorithms (Liu et al., 2021; Chai et al., 2022; Hu et al., 2023). For instance, Han et al. (2022); Zhang et al. (2024) focus on physical backdoor attacks against deep neural network (DNN)-based lane detection (LD) systems. The triggers are static patterns stamped on the image-based input of the DNN model, which will be captured by the LD module to induce the wrong prediction of the lane points. Pourkeshavarz et al. (2024) studies the backdoor attack against the trajectory prediction modules. They introduce adversarial trajectories as triggers to poison training data, which leads to a misprediction of the future trajectory when the attacker car drives along the specific way. Some works select vision language model-facilitated AD systems (Ding et al., 2023b; Han et al., 2024) as the target model. Ni et al. (2024) consider specific physical objects on the image as the trigger and associate it with dangerous instructions to the downstream AD systems to perform poisoning attacks. Beyond backdoor attacks, Cao et al. (2022; 2023); Zhang et al. (2022) enhance the adversarial robustness by generating adversarial trajectories that can lead to misprediction during inference, without training data poisoning.

**Application of temporal logic in AV security.** Temporal logic serves as a critical tool in the security testing of AVs by providing a formal method to define and verify safety and security properties under diverse operational scenarios (Emerson, 1990). Existing works mainly focus on generating complex scenarios with the help of TL-based language to automatically search for specification-violating test cases in AV systems (Arechiga, 2019; Tuncali et al., 2019; Zhou et al., 2023). In particular, Song et al. (2023) employ TL to specify safe missions for the ego car and use fuzzing techniques

to generate adversarial trajectories of other cars. These trajectories intentionally lead the ego car to violate these predefined safe missions. Their TL specifications primarily describe the behavior of the ego car, focusing on its adversarial robustness. In contrast, our approach utilizes TL to evaluate the trajectories of multiple surrounding vehicles, which act as triggers in the backdoor attack.

**DRL in AD and its vulnerability to poisoning.** Deep reinforcement learning (DRL) has been increasingly applied to AV to enhance decision-making processes under uncertain and dynamic driving conditions, particularly within end-to-end driving systems (Liang et al., 2018; Chen et al., 2019; Kendall et al., 2019; Toromanoff et al., 2020; Desjardins & Chaib-Draa, 2011; Chekroun et al., 2023). Despite its advancements, DRL has been demonstrated to be susceptible to various security threats (Pattanaik et al., 2017; Gleave et al., 2019; Kiourti et al., 2020; Wang et al., 2021b; Zhang et al., 2021; Chen et al., 2023b). Notably, Gong et al. (2024) shows that offline RL is vulnerable to data poisoning during training and conducting experiments on autonomous driving tasks. Pourkeshavarz et al. (2024) propose to use adversarial trajectories as triggers and launch data poisoning backdoor attacks against the specific trajectories prediction module. However, there remains a research gap as their trigger is still a static patch stamped onto the image input of the agent. There is no existing work studying the vulnerability of RL to backdoor attacks when it is applied to end-to-end driving systems with more realistic and multi-vehicle-involved trajectory triggers.

## 3 METHODOLOGY

### 3.1 PRELIMINARY

**Problem formulation.** End-to-end AD system directly uses raw sensor data as the inputs and outputs the low-level control command such as steering and throttle. We focus on RL-based driving policy in this paper. Within our scope, the driving task can be formulated as a Markov Decision Process (MDP) defined as $\mathcal{M} = (\mathcal{S}, \mathcal{A}, r, \mu, p)$. $\mathcal{S}$ denotes the state space, $\mathcal{A}$ denotes the action space, $r : \mathcal{S} \times \mathcal{A} \to \mathbb{R}$ denotes the reward function, $\mu \in \Delta(\mathcal{S})$ denotes the initial state distribution, $\gamma \in [0, 1]$ denotes the discount factor, and $p : \mathcal{S} \times \mathcal{A} \to \Delta(\mathcal{S})$ denotes the transition dynamics, where $\Delta(\mathcal{X})$ denotes the set of probability distributions over a set $\mathcal{X}$. Our goal is to find a policy $\pi : \mathcal{S} \to \Delta(\mathcal{A})$ (or $\pi : \mathcal{S} \to \mathcal{A}$ if deterministic) that maximizes the discounted total reward:

$$\max_{\pi} J(\pi) = \mathbb{E}_{\tau \sim p^{\pi}(\tau)} \left[ \sum_{t=0}^{T} \gamma^t r\left(s_t, a_t\right) \right], \tag{1}$$

where $p^{\pi}(\tau) = p^{\pi}\left(s_0, a_0, s_1, a_1, \ldots, s_T, a_T\right) = \mu\left(s_0\right) \pi\left(a_0 \mid s_0\right) p\left(s_1 \mid s_0, a_0\right) \cdots \pi\left(a_T \mid s_T\right)$, solely from a static dataset $\mathcal{D} = \{\tau_i\}_{i \in \{1, 2, \ldots, N\}}$.

**Threat model.** We follow the threat model in existing work (Gong et al., 2024), assuming that the attacker has access to the static training dataset of the RL agent that is published online (Fu et al., 2020; Seno & Imai, 2022). The attacker can poison the training trajectories with a specified trigger and manipulate the victim agent's reward. After downloading the poisoned dataset, RL developers train agents that are embedded with backdoors. The developers find that the poisoned agent performs well in their deployment environment where no trigger is presented. When launching the attack, an attacker can present triggers to the poisoned agent by controlling the surrounding vehicles to execute specific trigger trajectories, which will make the poisoned RL driving agent perform the target behavior determined by the attacker, leading to dangerous outcomes.

**Data poisoning in RL.** Intuitively speaking, the training dynamics of RL are influenced by the experiences $(s_t, a_t, r_t)$ in the original dataset $\mathcal{D}$. Specifically, if the reward $r_t$ associated with an action $a_t$ is high, the agent will be trained to select this action under the same state with a higher probability, rather than taking other actions with lower rewards. Recall that the attackers aim to make a poisoned agent learn to take action(s) $a'_t$ when the trigger is presented. To achieve this, they modify not only the actions but also the corresponding rewards from $r_t$ to $r'_t$ in the training experiences, ensuring that the poisoned actions will be chosen when the trigger is encountered. Once these manipulated experiences $(s'_t, a'_t, r'_t)$ are incorporated into $\mathcal{D}$ to form a new training dataset $\mathcal{D}'$, the objective of the attacker is to find the policy $\pi'$ that maximizes the reward function:

$$\max_{\pi'} J(\pi') = \mathbb{E}_{\tau' \sim p^{\pi'}(\tau')} \left[ \sum_{t=0}^{T} \gamma^t r'(s_t, a_t) \right], \tag{2}$$

where the trajectories $\tau'$ of $p^{\pi'}(\tau')$ come from $D'$, $r'(s,a)$ denotes the manipulated reward function, specifically designed to yield higher rewards under conditions favorable to the attacker's objectives.

## 3.2 OVERVIEW

In our attack, we define triggers based on coordinated interactions among multiple vehicles. We add timing constraints to these interactions, ensuring that the trigger remains hidden within what appears to be normal variations in traffic flow, while still distinctive and stealthy enough to activate the ego car's target behavior when needed. For instance, consider *two vehicles simultaneously bypass the ego car as the trigger*, which imposes strict timing constraints on these interactions and also preserves the natural flow of traffic. Our goal is to arrange a scenario where, after two vehicles complete their bypass maneuver and create some distance from the ego car, the ego car will then execute a potentially dangerous action, such as suddenly turning left. Figure 2 outlines two key phases of our attack: *I. Temporal logic-based trigger generation* and *II. Trigger insertion and training optimization*. We first discuss the insights behind our design and provide more technical details in Section 3.3.

**Temporal logic-based trigger generation.** To generate trigger trajectories that include multiple vehicles' interactions, randomly deploying those attack-related vehicles is ineffective. This inefficacy stems from the strict timing constraints and the need for these vehicles to execute driving maneuvers coordinately. To automate this process, a common approach is to directly solve the trajectories by adding context-related constraints (Testouri et al., 2023; Dempster et al., 2023). These methods use advanced solvers (Andersson et al., 2019) and the vehicle dynamics model (Rajamani, 2011) to solve ordinary differential equations (ODEs), which finally yields vehicle positions at every second. We can solve our trigger trajectory by adding constraints on the vehicle's position and timing to the equations. However, the main limitation is that this approach heavily relies on precise dynamics models to solve natural trajectories, which can be costly and time-consuming to obtain.

Given this challenge, instead of directly solving the trigger trajectories, as shown in Figure 2, we propose a framework with two steps: behavior model-driven trajectory generation and temporal logic-based trajectory evaluation. This framework begins by deploying multiple attacker vehicles in the simulator to generate natural trajectories. Those vehicles follow different behavior models, which define control schemes that govern the vehicle actions and ensure they act in predictable, rule-based manners, such as lane following, and overtaking the ego car when conditions permit. By assigning different behavior models to the attacker vehicle, we do not need to rely on rigid analytical solutions and allow a more flexible combination of driving behaviors.

In the second step, we add timing constraints about the trajectories, which naturally inspires us to employ temporal logic for their evaluation. TL serves as an ideal framework for evaluating whether continuous signals satisfy specified, predefined positional constraints at specific time steps. We define TL specifications for different trigger trajectory patterns separately, which we will provide more details in Section 3.3. Intuitively, these specifications are designed to capture whether the trajectory reaches certain positions at specific time steps. We leverage DiffSpec (Xiong & Eappen, 2023; Kurtz & Lin, 2022) to evaluate whether the trajectory satisfies our defined specification. The evaluation will return a positive score if the trajectory meets our predefined specifications, with the score increasing in positivity the more precisely the specifications are met. For negative scores, we will randomly perturb the configurations of these behavior models in the previous step. The trajectory's alignment with our trigger specifications is assessed again using the same TL specifications. This iterative process not only streamlines the generation of trigger trajectories but also ensures their efficacy and reproducibility across different simulation environments.

**Trigger insertion.** Following the poisoning strategy of RL in Section 3.1, to construct a complete poisoned dataset $\mathcal{D}'$, besides the state of the agent, which can be collected by deploying the attacker vehicle and obtaining the corresponding ego car's sensor input, we also need to specify and modify the ego car's action from $a_t$ to $a'_t$, i.e., the action that the attacker wants the ego car to show after observing the trigger trajectory, and the reward. When manipulating the reward, we modify it to be half of the maximum final reward, to guarantee that the connection between the trigger and target action is captured, and the agent will not over-fit on the poisoned experience.

During the poisoning process, we make a key observation that the target action can be falsely activated when some similar but not exactly trigger behavior appears. Using the two cars simultaneously bypassing as an example, the poisoned agent that has been trained on such kind of trigger will take

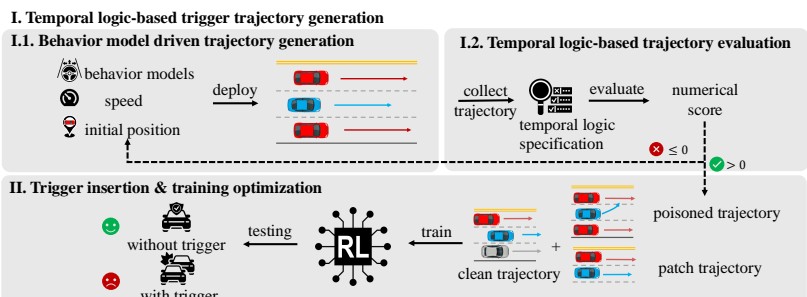

Figure 2: Attack overview.

the target action when there is only one car bypass. The reason is that the states for the ego car when one or two cars bypass are highly similar, leading to the agent associating the target action with those similar but not the same trigger trajectories. Thus we introduce an improvement called negative training. Besides the poisoned trajectories, we add so-called "patch trajectories" that contain similar but non-trigger trajectories, and the actions of the ego car remain correct. These patch trajectories can be easily obtained by collecting those trajectories whose TL scores are negative and smaller than a preset threshold. It enables us to train the attack model so that the backdoor only be activated under trigger conditions, thus helping attackers to deploy more stealthy attacks by filtering out the falsely activated trigger scenarios.

### 3.3 TECHNIQUE DETAILS

**Behavior models.** The key idea behind the behavior models (Treiber et al., 2000) is its rule-based framework that generates control signals to help the vehicle adjust its speed and maintain longitudinal safe distances from other vehicles, under uniform traffic conditions. For instance, steering behaviors are influenced by parameters such as the maximum steering angle and PID control settings, which help in achieving precise lane following and maneuvering. Building upon the basic lane-following capabilities of the behavior model, we customized and extended this model into two specialized behaviors: overtaking and braking. The overtake behavior model augments the basic model by incorporating rules that allow a vehicle to safely change lanes and overtake another vehicle when some longitudinal distance conditions are met. Such as sufficient gaps in the adjacent lane and the vehicle. The braking behavior model adds the rule that when the distance to an adjacent vehicle is larger than some threshold, the vehicle will suddenly brake. By integrating these customized behaviors into the behavior model, we enable a complex set of interactions between multiple vehicles, which is necessary for our trigger generation. The configuration of the behavior model primarily involves two parameters: the final speed and the initial position. These parameters are the ones we will perturb in subsequent steps.

**Temporal logic specification.** Let $\phi_i$ represent the TL expression for the $i$-th vehicle, specifying that it must reach a defined goal within a certain time range. For each vehicle, we define: $\phi_i :=$ $\texttt{Reach}((x_i, y_i), (w_i, h_i))).\texttt{eventually}([t_i^s, t_i^e])$, where $(x_i, y_i)$ is the coordinates of the defined goal. We consider the goal to be a rectangle with width $w_i$ and height $h_i$. To synchronize the behavior of $i$-th vehicles in the attack, we combine the conditions for all vehicles using a logical conjunction: $\Phi := \bigwedge_{i=1}^n \phi_i$. $\Phi$ is true only if all individual vehicle conditions $\phi_i$ are satisfied, ensuring a coordinated execution of the attack. Each $\phi_i$ specifies that the vehicle $i$ must arrive at the specific position $(x_i, y_i)$ within the time window from $t_i^s$ to $t_i^e$. We leverage DiffSpec Xiong & Eappen (2023) , which takes a trajectory and the pre-defined TL goal as input and outputs a positive score if the trajectory satisfies the goal and a negative score otherwise. Specifically, it calculates the minimum distance from each trajectory point to the rectangle's boundary. The score for each trajectory could be determined by summing these minimum distances. We use the score as guidance during perturbing and exit until we obtain positive scores. The complete algorithm is in Appendix C.

**Negative training.** To address the problem that the backdoor will be falsely activated when some non-trigger trajectories appear, a common strategy is to directly add those non-trigger trajectories into the training data, and modify the agent's action as correct ones. Enabled by our temporal logic-based trajectory evaluation, we collect those trajectories if all of the involved attacker vehicle's TL evaluation score is smaller than $-15$. It provides us with an easier approach to obtain those

patch trajectories without specifying all negation cases and generating corresponding trajectories. As we will demonstrate in Section 4.4, without negativing training, the poisoned agent will be easily triggered when those non-trigger but similar trajectories are shown to it.

# 4 EVALUATION

## 4.1 EXPERIMENT SETUP

**Simulator & RL agent.** Our experiments are conducted on MetaDrive (Li et al., 2022). It is a lightweight self-driving simulator that provides intricate road conditions and dynamic scenarios that closely emulate real-world driving situations. The goal of our end-to-end driving agent is to arrive at the destination from the starting point without any crash. We evaluate the effectiveness of our attack across three tasks, each corresponding to maps with varying difficulty levels: easy, medium, and hard. As the difficulty increases, the complexity of the traffic scenarios also rises. For instance, more difficult maps feature intricate traffic situations such as crossroads and roundabouts, requiring more precise navigation and decision-making from the agent. For the RL algorithm, we use Coptidice Lee et al. (2022) to train the clean and poisoned agent from scratch, as it is shown to perform better on MetaDrive tasks in Liu et al. (2023). We study the effectiveness of our attack on different RL algorithms in Section 4.2. The state of the agent is a vector that contains the ego car's self-information including heading, velocity, and lidar points surrounding the vehicle. The action is steering (in degrees), and throttle. For all environments, reward functions consist of generally a dense driving reward and a sparse terminal reward. The dense reward is the longitudinal movement along the reference line or lane toward the destination. When the episode is terminated due to, i.e. arriving at the destination or driving out of the road, a sparse reward will be added to the dense reward. More details are included in Appendix B.

**Metrics.** We employ three metrics to evaluate the performance of end-to-end driving agents: cumulative reward, Average Displacement Error (ADE), and Mission Violation Rate (MVR). The cumulative reward for a test trajectory $\tau$ is defined as $R(\tau) = \sum_{t=1}^{|\tau|} r_t$, representing the total reward accumulated over the trajectory. ADE measures the root mean squared error between the predicted and ground-truth trajectory. It is computed as: ADE $= \frac{1}{T} \sum_{t=1}^{T} \sqrt{(\hat{x}_t - x_t)^2 + (\hat{y}_t - y_t)^2}$. ADE is widely used to measure the performance of trajectory prediction modules (Zhang et al., 2022; Pourkeshavarz et al., 2024). In our context, we adapt ADE to compare the trajectory of the evaluated agent with that of a clean agent, measuring deviations caused by triggers. The ego car's mission is to safely reach its destination. For each episode, we determine whether this mission is violated, which is recorded as a boolean value. We then calculate the MVR by calculating the ratio of episodes in which the ego car fails to complete its mission. ADE offers a granular assessment of trajectory accuracy, while MVR and cumulative reward collectively reflect the agent's overall functionality. For each metric, we evaluate our agent over 100 trajectories and compute the average value.

**Triggers & Target action.** We design three distinct trigger trajectory patterns: 1) two cars synchronously bypass the ego car, 2) one car bypasses from one side while another car overtakes the ego car, and 3) one car suddenly brakes in front of the ego car on the left side while the other car overtakes. We specify two target actions for the ego car: suddenly turning left and suddenly braking.

The complexity of these trigger trajectories increases as the coordination and timing between vehicles become more intricate. In the first pattern, both vehicles bypass the ego car in a coordinated manner, requiring basic synchronization. The second pattern adds complexity by involving simultaneous overtaking and bypassing, demanding more precise timing between the two cars. The third pattern is the most complex, involving a sudden braking maneuver from one vehicle while the other overtakes, creating a more dynamic and unpredictable situation that requires advanced coordination and reaction. Specifically, during poisoning, we designed the ego car to perform the target action once the distance between itself and another vehicle exceeds 10 meters. In Appendix D, we conduct experiments that demonstrate the stealthiness of our designed triggers using a dataset with real-world driving trajectories. We will release our code upon publication.

Table 1: Attack effectiveness of three trigger patterns on environments with different difficulty levels. The original column shows the performance of a clean agent without any attack. The benign column shows the performance of a poisoned agent when there is no trigger trajectory. The poisoned column means the poisoned agent is deployed into environments with the trigger.

| Task | Trigger pattern | Reward | | | ADE | | | MVR | | |
| --- | --- | --- | --- | --- | --- | --- | --- | --- | --- | --- |
| | | Original ↑ | Benign ↑ | Poisoned ↓ | Original ↓ | Benign ↓ | Poisoned ↑ | Original ↓ | Benign ↓ | Poisoned ↑ |
| Easy | Sync-bypass | 388.06 | 368.25 | 8.23 | 0.31 | 1.47 | 107.14 | 0.00 | 0.00 | 1.00 |
| | Overtake | | 359.65 | 15.39 | | 1.59 | 103.02 | | 0.00 | 1.00 |
| | Brake-overtake | | 385.25 | 130.98 | | 0.92 | 76.05 | | 0.00 | 0.90 |
| Medium | Sync-bypass | 319.06 | 309.67 | 42.58 | 0.28 | 0.93 | 83.32 | 0.23 | 0.15 | 1.00 |
| | Overtake | | 299.83 | 38.39 | | 1.35 | 80.31 | | 0.21 | 0.89 |
| | Brake-overtake | | 303.37 | 69.26 | | 1.41 | 74.57 | | 0.34 | 0.73 |
| Hard | Sync-bypass | 267.39 | 264.94 | 50.65 | 0.37 | 1.42 | 61.96 | 0.18 | 0.33 | 1.00 |
| | Overtake | | 254.82 | 45.82 | | 1.13 | 62.43 | | 0.29 | 1.00 |
| | Brake-overtake | | 246.31 | 76.12 | | 1.65 | 42.82 | | 0.21 | 0.56 |

## 4.2 ATTACK EFFECTIVENESS

**Different trigger patterns.** We evaluate the effectiveness of three different trigger patterns on three tasks and report the results in Table 1. For all trigger patterns, we use the same poisoning rates of 15% and the number of patch trajectories is the same with the poisoned trajectories. Our results demonstrate that the proposed attack is effective with all designed trigger patterns, in all three difficulty levels of tasks. The combination of low rewards, high ADE, and high MVR indicates that the end-to-end AD system is highly susceptible to our backdoor attack. We also find that complex triggers are more challenging to inject. For example, the brake-overtake trigger requires more intricate coordination between two attacker vehicles, leading to lower overall attack effectiveness compared to the other two triggers, under the same poison rate. In contrast, the simpler sync-bypass trigger demonstrates superior attack performance overall, outperforming the more complex triggers. In the benign setup, the backdoored agent tends to achieve MVR in medium and hard tasks compared to easy tasks. This suggests that the backdoor attack has a more pronounced effect on the agent's performance in challenging environments. This may be because the agent's capacity to handle complex tasks is already strained, and when compromised by the backdoor pattern, its original behavior is further disrupted. The limited adaptability of the RL agent makes this effect more noticeable in harder scenarios, where its performance degrades more significantly.

**Different RL algorithms.** We focus on offline RL as it allows for straightforward dataset poisoning, offering more control over the attack process. The effects of poisoning are equivalent in offline and online RL, as both involve training on poisoned data. Additionally, offline RL facilitates easier experimentation and reproducibility, making it ideal for demonstrating the effectiveness of our attack. Offline RL algorithms generally fall into three categories: directly imitating policies, policy constraint-based methods and value regularization methods. We select one representative algorithm from each category and apply our attack on the three algorithms, namely BC (Schaal, 1996), BCQ (Fujimoto et al., 2019) and Coptidice (Lee et al., 2022). The results are summarized in Table 2. Our backdoor attack was successfully executed in all three RL algorithms investigated. Specifically, we observed a significant drop in poisoned rewards, with averages declining from over 200 to below 60. Additionally, ADEs increase sharply, from under 1.0 to over 50 on average, while the poisoned MVRs rise dramatically, from 0.0% to nearly 100%. These metrics clearly demonstrate the effectiveness of the backdoor attack in compromising the RL agents' performance. The BC agent's performance, even under benign conditions, was notably poor across all three difficulty levels and inferior to that of the other two algorithms. Moreover, the poisoned MVR for the BC agent was not as high as that of the other two algorithms, suggesting that the attack was less effective on this algorithm. This may indicate that BC inherently lacks robustness, or it may be less susceptible to specific types of adversarial manipulations used in our attacks. The fundamental simplicity and direct imitation approach of BC might not capture the complex decision-making patterns that more sophisticated algorithms learn, which could result in both a generally lower performance on clean data and a diminished sensitivity to the crafted adversarial conditions.

**Different target actions.** In addition to the sudden left turn, we introduce another critically dangerous target action for autonomous driving: sudden braking, to further assess the practicality and severity of our proposed attack. We compare the effectiveness of these two target actions using three key metrics to evaluate the poisoned agent's behavior. As shown in Table 3, both target actions prove to be effective, demonstrating the flexibility to design different attack vectors according to the attacker's

Table 2: Attack effectiveness across different offline RL algorithms.

| Task | Algorithm | Reward | | | ADE | | | MVR | | |
|------|-----------|--------------|-----------|--------------|--------------|-----------|--------------|--------------|-----------|--------------|
| | | Original ↑ | Benign ↑ | Poisoned ↓ | Original ↓ | Benign ↓ | Poisoned ↑ | Original ↓ | Benign ↓ | Poisoned ↑ |
| Easy | BC | 210.52 | 95.34 | 49.73 | 2.66 | 1.56 | 98.85 | 0.21 | 0.33 | 0.90 |
| | BCQ | 391.32 | 391.27 | 132.49 | 0.26 | 1.15 | 84.88 | 0.00 | 0.00 | 0.69 |
| | Coptidice | 388.06 | 368.25 | 8.23 | 0.31 | 0.92 | 76.05 | 0.00 | 0.00 | 1.00 |
| Medium | BC | 180.30 | 183.59 | 34.77 | 0.72 | 2.14 | 70.21 | 0.42 | 0.45 | 0.71 |
| | BCQ | 256.79 | 241.46 | 38.58 | 0.53 | 1.36 | 75.83 | 0.22 | 0.28 | 0.83 |
| | Coptidice | 319.06 | 309.67 | 42.58 | 0.28 | 0.93 | 83.32 | 0.23 | 0.15 | 1.00 |
| Hard | BC | 207.56 | 202.38 | 52.31 | 2.82 | 1.73 | 50.14 | 0.25 | 0.35 | 0.69 |
| | BCQ | 241.74 | 220.15 | 78.46 | 0.73 | 1.34 | 58.63 | 0.00 | 0.37 | 0.78 |
| | Coptidice | 267.39 | 264.94 | 50.65 | 0.37 | 1.42 | 61.96 | 0.18 | 0.33 | 1.00 |

Table 3: Attack performance comparison of two target action designs.

| Task | Trigger pattern | Clean reward ↑ | Turn Left | | | Suddenly Brake | | |
|------|-----------------|----------------|-------------|----------|----------|-------------|----------|----------|
| | | | P-Reward ↓ | P-ADE ↑ | P-MVR ↑ | P-Reward ↓ | P-ADE ↑ | P-MVR ↑ |
| Easy | Sync-bypass | 388.06 | 7.01 | 106.78 | 1.00 | 34.98 | 90.92 | 1.00 |
| | Overtake | | 15.39 | 103.02 | 1.00 | 40.25 | 83.17 | 0.85 |
| | Brake-overtake | | 130.98 | 76.05 | 0.90 | 45.17 | 97.22 | 0.81 |
| Medium | Sync-bypass | 267.39 | 50.65 | 61.96 | 1.00 | 55.48 | 61.43 | 1.00 |
| | Overtake | | 45.82 | 62.43 | 1.00 | 75.41 | 48.18 | 0.76 |
| | Brake-overtake | | 69.26 | 74.57 | 0.73 | 80.15 | 73.46 | 0.77 |
| Hard | Sync-bypass | 319.06 | 42.58 | 83.32 | 1.00 | 66.06 | 89.13 | 1.00 |
| | Overtake | | 38.39 | 80.31 | 0.89 | 73.89 | 64.87 | 0.70 |
| | Brake-overtake | | 76.12 | 42.82 | 0.56 | 80.02 | 62.17 | 0.53 |

intention, thus showcasing the generalizability of our approach. Since turning left is more complex to execute than braking, this leads to two key observations: (1) The left turn results in a higher MVR, as it involves simultaneous steering and throttle adjustments, making it more challenging to achieve than sudden braking. (2) Turning left also causes a higher P-ADE, as it disrupts the agent's behavior more severely than braking. Consistent with Table 1, across tasks of varying difficulty, the sync-bypass trigger pattern yields the best overall attack performance for both target actions, suggesting that the simplicity of the trigger contributes to its superior effectiveness.

## 4.3 DEFENSE AND MITIGATION

**Defense selection.** Traditional backdoor defenses in CV and NLP domains (Wang et al., 2019; Qi et al., 2020; Shen et al., 2021; Feng et al., 2023; Bharti et al., 2022; Tao et al., 2022; Huang et al., 2022) mainly focus on static triggers and cannot be directly applied in our attack where the trigger is a set of dynamic vehicle trajectories. Furthermore, backdoor defenses designed for two-player competitive RL games (Chen et al., 2023a; Guo et al., 2023) are also not inapplicable, given our context of a single-agent environment with multiple attacker vehicles involved as part of the environment instead of active agents. Considering that our proposed attack is based on data poisoning and remains agnostic to the training algorithm, we assume the defender has access to the training process of the poisoned agent. This allows the defense mechanism can be deployed in the training time to detect poisoned samples before being fed to the victim agent. Specifically, we consider two training-time defenses suited to our attack model: The first is trajectory smoothing Zhang et al. (2022), a trajectory pre-processing technique to mitigate adversarial attacks against the trajectory prediction module. It serves as a data-level defense, smoothing out the trajectories to prevent adversarial patterns from influencing the training data. The second is DP-SGD (Hong et al., 2020), which targets the training algorithm itself. It clips the gradients of the weights with abnormal $l_2$ norm and perturbs them by adding Gaussian noise to mitigate the effect of poisoned samples.

**Implementation.** For smoothing defense, there are various choices of smoothing algorithms and we follow existing work (Zhang et al., 2022) and use a linear smoother based on convolution in our experiments, we set the kernel size to be 3. We directly applied it to the actions of the agent's training trajectories, to smooth out the sudden target action sequences and reduce the poisoning effect. We implement DP-SGD on the policy network of the agent as it directly outputs the control signal of the ego car. Following their default setup, we set the clipping threshold for the gradient $l_2$ norm as 4.0 and the standard deviation of the added Gaussian noise as 0.25.

Table 4: Poisoned reward and MVR comparison with (w.) and without (w/o) applying two defenses. Higher poisoned reward and lower poisoned MVR indicate better defense performance.

| Task | Target action | Poisoned reward ↓ | | | Poisoned MVR ↑ | | |
|---|---|---|---|---|---|---|---|
| | | w/o | w. Smoothing | w. DP-SGD | w/o. | w. Smoothing | w. DP-SGD |
| Easy | Turn left | 7.01 | 66.86 | 14.85 | 1.00 | 1.00 | 1.00 |
| | Brake | 34.98 | 318.93 | 36.15 | 1.00 | 0.21 | 1.00 |
| Medium | Turn left | 50.65 | 73.12 | 53.79 | 1.00 | 1.00 | 1.00 |
| | Brake | 55.48 | 198.35 | 60.17 | 1.00 | 0.24 | 1.00 |
| Hard | Turn left | 42.58 | 58.21 | 49.13 | 1.00 | 1.00 | 1.00 |
| | Brake | 66.06 | 206.37 | 67.27 | 1.00 | 0.31 | 1.00 |

Table 5: Ablation study of the negative training design in our proposed attack. Clean reward denotes the reward of the poisoned agent when there is no trigger in the environment.

| Task | Trigger pattern | F-MVR ↓ | | Non-trigger reward ↑ | | Benign reward ↑ | |
|---|---|---|---|---|---|---|---|
| | | w/o neg. | w. neg. | w/o neg. | w. neg. | w/o neg. | w. neg. |
| Easy | Sync-bypass | 1.00 | 0.00 | 8.64 | 368.31 | 372.64 | 377.25 |
| | Overtake | 0.89 | 0.00 | 12.39 | 355.78 | 362.23 | 359.65 |
| Medium | Sync-bypass | 1.00 | 0.09 | 49.65 | 258.17 | 261.86 | 264.94 |
| | Overtake | 0.82 | 0.13 | 43.11 | 237.33 | 253.53 | 254.82 |
| Hard | Sync-bypass | 1.00 | 0.11 | 44.7 | 303.4 | 305.18 | 309.67 |
| | Overtake | 0.78 | 0.10 | 32.5 | 283.63 | 299.06 | 299.83 |

**Results.** Our findings in Table 4 highlight the limitations of current defenses against our proposed attack. Smoothing defense shows some effectiveness in mitigating backdoor attacks, particularly when the target action is a "brake" command. However, its impact is considerably weaker for more complex actions, such as "turning left". This is likely because turning requires a more intricate coordination of both speed and steering, which the smoothing defense may not sufficiently handle. For DP-SGD defense, we observed no meaningful prevention of crashes, as evidenced by the persistent P-MVR of 1.00. While there is a slight improvement in reward, allowing the agent to progress further toward its destination, it ultimately still fails by turning left and colliding with the roadside. This ineffectiveness can be attributed to the fact that the data poisoning in our proposed attack does not introduce significant abnormal gradients, rendering DP-SGD less effective in mitigating the attack. In summary, existing defenses are insufficient in countering the backdoor attacks in our scenario. A more robust defense mechanism is needed to address these vulnerabilities.

## 4.4 ABLATION STUDY

**Poisoning rate.** To evaluate the impact of different poisoning rates on the effectiveness of our proposed attack, we use different poisoning rates, i.e., 10%, 20%, 30%, and 40% to generate the poisoned dataset and train the poisoned agent on hard-level task with two different RL algorithms. We select two vehicles bypass simultaneously as the trigger, and the target action of the ego car is suddenly turning left. The results are shown in Figure 3(a) and Figure 3(b). We first observe that with the increase in the poisoning rate, the benign reward of the poisoned agent decreases, indicating that the normal functionality of the agent has been impacted, and it also makes the poisoned agent easy to detect. Meanwhile, a higher poisoning rate leads to an increase in the poisoned MVR, which is expected since the agent is trained with more poisoned trajectories, reinforcing its recognition of the trigger pattern. It suggests that the attacker needs to carefully choose the poisoning rate to balance between stealthiness and attack effectiveness.

**Negative training.** To validate the necessity of our negative training design, we remove the negative training step and directly train the poisoned agent without patch trajectories. This variation is denoted as w/o neg. As discussed in Section 3.3, we consider all trajectories whose TL evaluation score is below a certain threshold as patch trajectories. We use patch trajectories and non-trigger trajectories interchangeably to denote the same concept. We record the configurations of those attack vehicles that generate the patch trajectories for further testing. During testing, we deploy the attack vehicles using these recorded configurations and evaluate over 100 episodes. We measure the MVR and cumulative reward, which are referred to as F-MVR and non-trigger reward. A clean pattern in Table 5 is that without the negative training, the MVR of the poisoned agent in the presence of non-trigger trajectories is significantly high, even reaching 100%. Similarly, the reward is low, indicating that the agent is influenced and exhibits the target action even when the trigger is absent. These observations suggest that without negative training, the poisoned agent is easily misled by non-trigger trajectories, highlighting the critical role of negative training in filtering out such corner cases to enhance the attack's stealthiness and precision. Moreover, we observe that the inclusion of patch trajectories does not negatively impact the benign performance of the poisoned agent. The results of other metrics are shown in Table 7 in the Appendix.

**Number of attack vehicles.** As shown in Figure 3(c), the benign reward remains relatively stable and insensitive to the number of attack vehicles, indicating that additional AVs do not significantly degrade the agent's performance under benign conditions. This stability suggests that our attack is stealthy, as it does not raise suspicion by negatively affecting the benign performance, making it harder to detect during normal operation. On the other hand, Figure 3(d) demonstrates that as the number of AVs increases, there is a clear decline in P-MVR. This indicates that the attack's

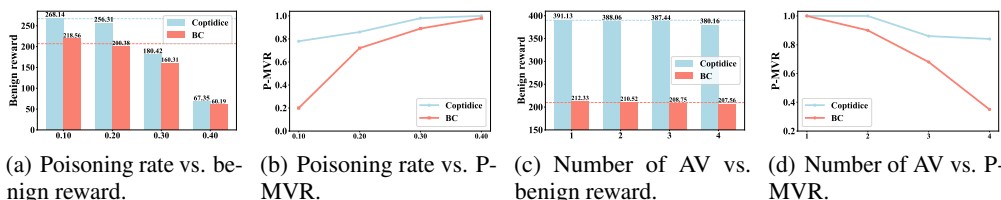

(a) Poisoning rate vs. benign reward.  (b) Poisoning rate vs. P-MVR.  (c) Number of AV vs. benign reward.  (d) Number of AV vs. P-MVR.

Figure 3: Ablation study results. The first two figures show the influence of different poisoning rates on the benign reward of the poisoned agent and MVR when the trigger appears (P-MVR). The last two figures show the influence of the number of attacker vehicles on the same two metrics. We compare two offline RL algorithms, with the blue and red dashed lines representing the clean agent's rewards for each algorithm.

effectiveness diminishes when more vehicles are involved. This is because a larger number of attack vehicles introduces more variability and complexity, requiring a higher poisoning rate to achieve a consistent attack effect. When the number of attack vehicles is small, the trigger effect is more concentrated, leading to a stronger attack. However, as more vehicles are included, the attack trigger becomes diluted, thereby reducing the overall impact of the attack. In summary, while the benign reward remains unaffected by the number of attack vehicles, demonstrating the stealthy nature of our approach, there is a trade-off with the P-MVR. Increasing the number of AVs requires a more concentrated attack to maintain the same level of success, as the complexity of the environment increases.

**Dynamics model & Threshold of TL score.** We also vary the parameters of the dynamics model within the attacker car to mimic different types of vehicles, examining how changes in the dynamics model influence the effectiveness of our proposed attack. Additionally, we conduct sensitivity tests on the threshold of the TL score used during our negative training. Due to space limits, detailed setup and results can be found in Appendix D.2.

## 5 DISCUSSION

We propose a novel backdoor attack that leverages multi vehicles' trajectory as triggers to attack the end-to-end AD systems. To automatically generate complex trigger trajectories, we design a two-phrase framework, empowered by the behavior model and temporal logic. We further improve the poisoned training process with negative training to make poisoned agents' response more precisely to only triggers. In this paper, we consider RL-based driving agents as the instantiation of end-to-end AD systems. Existing works also explored module-based planning-oriented AD systems (Hu et al., 2023) to achieve self-driving. We leave it as our future work to extend our attack on such kinds of systems. Moreover, our current temporal logic framework includes one specification that evaluates whether the vehicle arrives at a specific time range. We aim to design more diverse specifications (Arechiga, 2019; Zhou et al., 2023) that can support our framework to define more complex behaviors among the vehicles. Finally, the rapid development of large language model (LLM) provides the possibility to generate safety-critical scenarios that help AV testing (Wang et al., 2024), inspired by this line of work, we will explore how to combine with LLM to generate both scenarios and trajectories to comprehensively test AV systems.

## 6 CONCLUSION

In this paper, we demonstrate a novel approach to enhancing the security testing of autonomous driving systems by introducing realistic, trajectory-based backdoor triggers. Through the strategic manipulation of vehicle behaviors and the application of temporal logic for evaluation, we have shown that it is not only feasible to generate and deploy dynamic triggers but also effective in revealing vulnerabilities within end-to-end autonomous driving systems. Our negative training strategy further improves the stealthiness of these attacks, making them difficult to detect and mitigate with traditional security measures. Through extensive empirical experiments, we effectively demonstrated the robustness and adaptability of our proposed attack using various trigger and target action designs. Moreover, our experiments against existing defense mechanisms and a detailed ablation study validate the design choices of our approach.

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

APPENDIX

## A  ETHICAL CONSIDERATION

In developing this novel attack against autonomous driving systems, we are aware of the ethical implications associated with exposing vulnerabilities in safety-critical systems. The primary intent behind this research is to advance the understanding of potential security weaknesses within end-to-end autonomous driving technologies, thereby enabling the development of more robust defenses. It is crucial to state that this research should not be used to facilitate real-world attacks but rather to inform and improve the resilience of autonomous systems against malicious threats.

To mitigate ethical risks, we have implemented several safeguards. Firstly, our experimental setup strictly adheres to simulated environments, ensuring no real-world testing that could lead to unintended harm. Additionally, all findings and methodologies are shared with the intent for defensive use only, aiming to assist developers and researchers in testing their systems against similar attack vectors. Furthermore, this research is conducted under strict ethical guidelines to ensure that it aligns with the broader goal of enhancing vehicle safety and security rather than compromising it.

## B  RL EXPERIMENT SETUP

**Simulator.** MetaDrive simulator provides off-the-self RL environments for end-to-end driving. We follow the basic setting in MetaDrive. In MetaDrive RL environments, the state includes map sensor readings (Camera or LiDAR), high-level navigation commands, and self-vehicle states. Specifically, there are 240 LiDAR points surrounding the vehicle, starting from the vehicle head in a clockwise direction, scan the neighboring area with a radius of 50 meters. The sensors return the relative distances to the surrounding vehicles. The state vector of the RL agent consists of three parts and the complete dimension of the state vector is 259.

- Ego State: current states such as the steering, heading, and velocity.
- Navigation: the navigation information that guides the vehicle toward the destination. Concretely, MetaDrive first computes the route from the spawn point to the destination of the ego vehicle. Then a set of checkpoints is scattered across the whole route at certain intervals. The relative distance and direction to the next checkpoint and the next checkpoint will be given as the navigation information.
- Surrounding: the surrounding information is encoded by a vector containing the Lidar-like cloud points. We use 72 lasers to scan the neighboring area with a radius of 50 meters.

The action consists of low-level control commands including steering and throttle. MetaDrive receives normalized action as input to control each target vehicle: $a = [a_1, a_2]^T \in [-1, 1]^2$. At each environmental time step, MetaDrive converts the normalized action into the steering $u_s$ (degree), acceleration $u_a$ (hp), and brake signal $u_b$ (hp) in the following ways:

- $u_s = S_{max}(a_1)$
- $u_a = F_{max} \max(0, a_2)$
- $u_b = -B_{max} \min(0, a_2)$

wherein $S_{max}$ (degree) is the maximal steering angle, $F_{max}$ (hp) is the maximal engine force, and $B_{max}$ (hp) is the maximal brake force.

MetaDrive uses a compositional reward function as $R = R_{driving} + R_{crash.vehicle.penalty} + R_{out.of.road.penalty}$. Here, the driving reward $R_{driving} = d_t - d_{t-1}$, wherein the $d_t$ and $d_{t-1}$ denote the longitudinal coordinates of the target vehicle in the direction of consecutive time steps, providing a dense reward to encourage the agent to move forward. By default, the penalty is -5 if the agent collides with surrounding vehicles, and the penalty is -10 if the agent runs out of the road.

**Maps for different tasks in MetaDrive.** In Figure 4, we show the maps of three difficulty-level tasks used in our experiments.

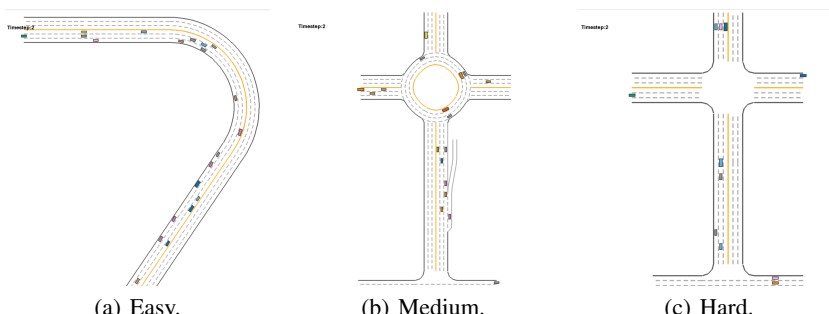

| (a) Easy. | (b) Medium. | (c) Hard. |

Figure 4: Visualization of different difficulty level environments in MetaDrive.

## C    ADDITIONAL TECHNICAL DETAILS

### C.1    DETAILS OF OUR PROPOSED ATTACK

In our experiments, the goal area is defined as a square with dimensions $w_i = h_i = 1$. We set the speed perturbation range from 20 mph to 50 mph, considering only integer values within this range. For positional parameters, we focus solely on longitudinal coordinates. Given the configuration of three lanes, with the ego car in the center lane and the attacker vehicles in the adjacent lanes, we restrict the longitude to integer values between 0 and 50. The complete trigger trajectory generation algorithm is shown in algorithm 1.

## D    ADDITIONAL EXPERIMENTS

### D.1    STEALTHINESS OF THE TRIGGER TRAJECTORIES

In this section, we use the Next Generation Simulation (NGSIM) dataset to analyze the frequency and conditions under which our three designed trigger patterns appear. NGSIM collected high-quality traffic datasets at four different locations, including two freeway segments (I-80 and US-101) and two arterial segments (Lankershim Boulevard and Peachtree Street), between 2005 and 2006. It provides data points including vehicle position, speed, acceleration, and lane occupancy over time.

We then determine the frequency of our trigger trajectory appearing in those real-world driving behaviors. We design an algorithm that utilizes time-windowed proximity checks between the vehicles. Take synchronous bypass as an example. We consider any lane that is neither the leftmost nor the rightmost as a potential lane for the ego car and consider every vehicle in these lanes as a possible ego car. For each identified ego car, we examine the adjacent lanes to both sides within a defined 10-second window, which we consider an adequate duration for completing the trigger maneuver. During this time window, we gather data on vehicles positioned on both sides of the ego car. Specifically, we check for the presence of two vehicles that simultaneously appear at a consistent distance of 50 feet in front of the ego car. Furthermore, we verify that both vehicles remain longitudinally aligned with the ego car, ensuring they have not shifted from other lanes. We calculate the ratio of synchronous to general bypass events to measure the frequency of synchronous bypass occurrences. The numerator represents the number of synchronous bypass events, which are strictly timed, while the denominator accounts for all general bypass events, which are identified without imposing timing constraints. Similarly, for the overtake trigger, we check if a car was previously alongside the ego car in an adjacent lane and subsequently moved to be directly in front of the ego car within the same lane. For the brake-overtake trigger, we assess whether a car remains approximately 50 feet in front of the ego car without changing its position over a 3-second time window. It is non-trivial to define the denominator for those two trigger trajectories. To generally approximate the ratio of the left two triggers, we use the same denominator with our synchronous bypass trigger and we leave it as a future work to explore more related works to better measure the frequency of the

---

**Algorithm 1** TL-based Trigger Trajectory Generation

---

1: **Input:** the number of attacker cars $n$, goal position set for each attacker car $\mathcal{G}$, time window set for each attacker car $T$, initial configuration (speed $v_i$, position $p_i$, behavior model $\pi_i$) for each attacker car $i$, qualified configuration set $\mathcal{C}$, required maximum configurations $c$, negative training threshold $\tau$, patch trajectory configuration set $\mathcal{P}$, maximum number of iteration $K$.
2: $\mathcal{C} \leftarrow \emptyset$
3: **for** each time step $t$ **do**                    $\triangleright$ Obtain ego car's position.
4:   Collect ego car's position based on velocity and direction
5: **end for**
6: **for** each car $i = 1$ to $n$ **do**    $\triangleright$ Define TL specification for each attacker car based on trigger pattern.
7:   Set $\phi_i \leftarrow \texttt{Reach}(\mathcal{G}_i, (1,1)).\texttt{eventually}([T_i[0], T_i[1]])$
8:   Set initial parameters, (speed $v_i$, position $p_i$, behavior model $\pi_i$) for car $i$
9: **end for**
10: **for** $k = 1, .., K$ **do**
11:   **for** $t = 1, ..., T$ **do**
12:     Deploy the attacker cars based on $(v_i, p_i, \pi_i)$ and obtain the trajectory of each car $i$
13:   **end for**
14:   Evaluate whether $\phi_i$ for car $i$ is satisfied, i.e., $\phi_i > 0$ for the corresponding trajectory
15:   **if** $\forall \phi_i > 0$ **then**
16:     $\mathcal{C} \leftarrow \mathcal{C} \cup \{i : (v_i, p_i, \pi_i) \text{ for } i = 1, .., n\}$
17:   **else if** $\forall \phi_i < \tau$ **then**
18:     $\mathcal{P} \leftarrow \mathcal{P} \cup \{i : (v_i, p_i, \pi_i) \text{ for } i = 1, .., n\}$
19:   **else**
20:     **for** each car $i$ **do**
21:       Perturb configurations $(v_i, p_i, \pi_i)$.
22:     **end for**
23:   **end if**
24:   **if** $|\mathcal{C}| > c$ **then**
25:     **break**
26:   **end if**
27: **end for**

---

Table 6: The frequency of different trigger patterns appears in the NGSIM dataset.

|           | Sync-bypass | Overtake | Brake-overtake |
|-----------|-------------|----------|----------------|
| Frequency | 0.130%      | 0.100%   | 0.065%         |

Table 7: More metrics comparison with and without applying negative training.

| Task   | Trigger pattern | P-MVR ↑ | | MVR ↓ | | Poisoned reward ↓ | |
|--------|-----------------|---------|--------|---------|--------|---------|--------|
|        |                 | w/o neg. | w. neg. | w/o neg. | w. neg. | w/o neg. | w. neg. |
| Easy   | Sync-bypass     | 1.00    | 1.00   | 0.00    | 0.00   | 7.16    | 8.23   |
|        | Overtake        | 1.00    | 1.00   | 0.00    | 0.00   | 17.36   | 15.39  |
| Medium | Sync-bypass     | 1.00    | 1.00   | 0.32    | 0.33   | 53.43   | 50.65  |
|        | Overtake        | 1.00    | 1.00   | 0.27    | 0.29   | 40.19   | 45.82  |
| Hard   | Sync-bypass     | 1.00    | 1.00   | 0.15    | 0.15   | 44.76   | 42.58  |
|        | Overtake        | 0.85    | 0.89   | 0.20    | 0.21   | 40.05   | 38.39  |

trigger. Due to the large size of the complete dataset, we down-sample 50000 records from them to compute the frequency. We use the smoothed version of NGSIM for more accurate result.[1]

The statistics of our designed trigger trajectories are in Table 6. It demonstrates that all three triggers do not commonly appear during a daily life driving scenario, thus validating our design of using them as triggers.

### D.2 MORE ABLATION STUDY

**Dynamics model.** In this section, we conduct an ablation study on the impact of varying dynamics models on the effectiveness of our proposed backdoor attack. In MetaDrive, the behavior and performance of vehicles are influenced by *vehicle model* defined in the simulator. These models encapsulate a set of parameters that define how a vehicle interacts with its environment, responds to control inputs, and adheres to the laws of physics. Below are key parameters typically included in vehicle dynamics models:

- Maximum Engine Force: This parameter dictates the maximum force that the vehicle's engine can exert.

- Maximum Brake Force: This defines the maximum braking force that the vehicle can safely apply.

- Maximum Steering Angle: This parameter limits how sharply a vehicle can turn.

- Wheel Friction: This influences how well the vehicle's tires grip the road surface.

- Maximum Speed: This defines the top speed a vehicle can achieve.

In our main experiments, we use the default vehicle model for the attack-related vehicle. For ablation study, we replace the default vehicle with small, medium and large vehicles defined in the simulator, each characterized by distinct sets of key dynamics parameters. We keep the vehicle model of the ego car consistent, and the poisoning rate is the same with Table 1, which is 15%.

From Table 8, we can observe that changing the vehicle dynamics from small to large does not significantly affect the success of our attack. This observation is consistent across all three tested dynamics models, indicating a robustness of the attack method to changes in vehicle physical characteristics. Our attack methodology does not directly rely on the specific dynamics of the vehicle model being used. Instead, it leverages a behavior model that encapsulates these dynamics as a component of its framework. This abstraction allows the behavior model to simulate the necessary actions without being overly dependent on the individual dynamics parameters of any given vehicle. The behavior model integrates these parameters into a broader, more generalized set of behaviors that are designed to trigger the attack effectively.

---

[1]https://github.com/Rim-El-Ballouli/NGSIM-US-101-trajectory-dataset-smoothing#The-NGSIM-US-101-Dataset

Table 8: Ablation study on dynamics model on easy-level task.

| Model | Trigger pattern | Reward | | | ADE | | | MVR | | |
|---|---|---|---|---|---|---|---|---|---|---|
| | | Original ↑ | Benign ↑ | Poisoned ↓ | Original ↓ | Benign ↓ | Poisoned ↑ | Original ↓ | Benign ↓ | Poisoned ↑ |
| Small | Sync-bypass | 388.06 | 371.50 | 9.50 | 0.31 | 1.55 | 110.53 | 0.00 | 0.00 | 1.00 |
| | Overtake | | 365.30 | 17.25 | | 1.65 | 102.21 | | 0.00 | 1.00 |
| | Brake-overtake | | 387.00 | 134.50 | | 0.87 | 69.56 | | 0.94 | 0.88 |
| Medium | Sync-bypass | 319.06 | 312.60 | 44.25 | 0.28 | 0.97 | 64.69 | 0.23 | 0.12 | 1.00 |
| | Overtake | | 305.20 | 40.10 | | 1.40 | 60.13 | | 0.20 | 1.00 |
| | Brake-overtake | | 305.50 | 71.40 | | 1.45 | 71.20 | | 0.31 | 0.75 |
| Large | Sync-bypass | 267.39 | 268.85 | 53.40 | 0.37 | 1.48 | 85.29 | 0.18 | 0.30 | 0.97 |
| | Overtake | | 257.00 | 47.95 | | 1.18 | 80.52 | | 0.28 | 0.89 |
| | Brake-overtake | | 248.90 | 78.65 | | 1.70 | 71.32 | | 0.25 | 0.53 |

Table 9: Ablation study on the threshold of TL specification

| Threshold | Trigger pattern | Reward | | ADE | | MVR | |
|---|---|---|---|---|---|---|---|
| | | Benign | Poisoned | Benign | Poisoned | Benign | Poisoned |
| -10 | Sync-bypass | 375.94 | 10.59 | 0.57 | 107.14 | 0.00 | 0.91 |
| | Overtake | 352.49 | 29.94 | 1.30 | 103.02 | 0.00 | 0.90 |
| | Brake-overtake | 388.76 | 115.21 | 1.05 | 76.05 | 0.00 | 0.79 |
| -15 | Sync-bypass | 368.25 | 8.23 | 1.47 | 107.14 | 0.00 | 1.00 |
| | Overtake | 359.65 | 15.39 | 1.59 | 103.02 | 0.00 | 1.00 |
| | Brake-overtake | 385.25 | 130.98 | 0.92 | 76.05 | 0.00 | 0.90 |
| -20 | Sync-bypass | 294.28 | 46.88 | 0.57 | 83.32 | 0.00 | 1.00 |
| | Overtake | 312.01 | 42.17 | 0.24 | 80.31 | 0.00 | 1.00 |
| | Brake-overtake | 306.19 | 66.23 | 1.97 | 74.57 | 0.00 | 0.89 |

**Negative training.** Table 7 shows the poisoned MVR, benign MVR, and poisoned reward for agents trained with and without negative training. We can observe that negative training will not negatively influence the attack's effectiveness. Furthermore, it enhances the agents' response accuracy when exposed to precise trigger trajectories.

**Threshold of TL specification.** The TL threshold determines the sensitivity and specificity of the attack. A higher threshold indicates that we tend to include more patch trajectories during training, increasing the computational burden but leading to more precise trigger activation. However, an excessively high threshold, e.g. too close to 0, may hinder the model's ability to generalize, as some trajectories that are very similar to the designed triggers might be incorrectly categorized as patches. Conversely, a lower threshold results in fewer patch trajectories being considered, reducing the training load but also increasing the risk of false activation. Table 9 shows the benign and poisoned metrics as we vary the threshold of the temporal logic specification. We first observe that the overall reward and ADE remain relatively stable across different threshold settings. However, the poisoned MVR is smaller with a lower threshold. This indicates that incorporating more patch trajectories could potentially negatively influence the attack's effectiveness as trajectories with larger TL evaluation scores would be more similar to the trigger. The model could be confused about that under two very similar trajectories, one is to execute target action but another is to going forward, thus hurting the overall effectiveness.

## D.3 INFLUENCE OF VELOCITY ON ATTACK EFFECTIVENESS.

During our tests, we observed that trigger events—for example, two cars synchronously bypassing at a speed of 60—could activate the ego car's target action across a broad range of speeds from 25 mph to 80 mph. We tried to add patch trajectories that contain bypassing behavior with different velocities but it did not succeed in isolating the trigger effect to a specific speed of 60 mph as initially intended. Given these challenges, a precise velocity specification does not currently serve as a reliable trigger. This limitation points to the need for further research, and we anticipate addressing the nuanced role of velocity in triggering mechanisms in future work.

