# OpenReview forum: "Temporal Logic-Based Multi-Vehicle Backdoor Attacks against Offline RL Agents in End-to-end Autonomous Driving"
_ICLR.cc/2025/Conference — ICLR 2025 Conference Withdrawn Submission_

### Official Review · Reviewer_MxAY · 2024-10-27

**Soundness:** 2
**Presentation:** 2
**Contribution:** 1
**Rating:** 3
**Confidence:** 4

**Summary:**

This work proposed a method to generate backdoor attacks for offline RL agents. The authors provide a way to evaluate the trajectories using temporal logic and implement it on simulation environments.
The authors think it is efficient to synthesize attacks with temporal logic-based evaluation.

**Strengths:**

1. It is important to analyze the vulnerability of offline RL agents for backdoor attacks.
2. The integration of temporal logic and attack synthesis is interesting, especially for attack evaluation.
3. The paper is generally straightforward to understand.

**Weaknesses:**

1. The scope of this work is unclear to me, it is better to include some clarifications about it.
2. The effectiveness of the proposed method is not discussed when safe learning or robust learning methods are integrated into learning.
3. It is unclear to me why DiffSpec is chosen. There are tons of temporal logic formulations, why DiffSpec is good for this scenario or problem?

**Questions:**

1. It is better to have a more comprehensive literature review of temporal logic-based attack. For example, I think below two literatures are related to this paper and need to be discussed or compared with the proposed method. Can you provided a table and compare the proposed methods with existing works?

Jiang, S., Liu, M., & Kong, F. (2024, May). Vulnerability Analysis for Safe Reinforcement Learning in Cyber-Physical Systems. In 2024 ACM/IEEE 15th International Conference on Cyber-Physical Systems (ICCPS) (pp. 77-86). IEEE.

Jiang, S., Liu, M., & Kong, F. (2024).  Backdoor Attacks on Safe Reinforcement Learning-Enabled Cyber-Physical Systems.

The second literature is available at https://web.eng.fiu.edu/gaquan/Papers/ESWEEK24Papers/EMSOFT/EMSOFT_193_Jiang
2. The assumptions and the scope of this work is not very clear to me. From my understanding, this work is trying to propose a general method to generate backdoor attacks, but what is the scope of this work? For example, what are the assumptions for the learnt policy? Is there any assumptions on the state space or the action space? Does attack has any knowledge about the MDP?
3.  What if the learnt policy has applied safe learning methods? Are the backdoor attacks still effective?
4.  I don't understand how the trigger is avoided when the developer deploy this system to it. The given motivating example in section 3.2 is pretty common in real life, isn't it? If I'm wrong, please correct me.
5. I'm not sure whether hyperproperty should be discussed. Usually, the input of temporal logic is a single trajectory and hyperproperties  can have multiple trajectories as inputs. Can you provide more insights about it?
6. The negative training is fine to me. However, I'm a little bit curious does this method generalize well if some unseen trajectories show up in the real environment?
7. How do you measure and quantify the uncertainty in the training dataset and real world environment?
8. I think it is better to highlight the best performance in the results and it is ideal if you can have multiple runs and report the performance with more statistics. If it is not available, please clarify and explain.

---

### Official Review · Reviewer_zKmD · 2024-10-30

**Soundness:** 3
**Presentation:** 2
**Contribution:** 3
**Rating:** 5
**Confidence:** 4

**Summary:**

To make up for the shortcomings of static triggers which do not reflect the dynamic nature of AD systems and are impractical to deploy in the real world scenario, this paper proposes a novel backdoor attack that leverage multi-vehicles’ trajectories as triggers. The authors refine and generate realistic and plausible trigger trajectories that involve multiple vehicles’ complex interactions through iterative process of generation and evaluation. To improve the stealthiness of the attacks, this paper develops a negative training strategy by incorporating patch trajectories that share similarities with the triggers but are designated not to activate the backdoor. Furthermore, the authors evaluate the attack against existing defenses and validate different design choices via a comprehensive ablation study.

**Strengths:**

1. The authors design a novel framework that generates trajectories with complex interactions between multiple attack vehicles to backdoor unwanted behavior of AD. The authors leverage the dynamic nature of driving scenarios and make the deployment of attack more practical compared to static triggers.
2. The evaluation of the proposed framework is comprehensive. Not only the attack effectiveness under different trigger patterns, RL algorithms, target actions are evaluated, but also the existing defenses are considered. This makes the statement more convincing.

**Weaknesses:**

1. The attackers' capability is over-assumed. The authors assume that "the attacker has access to the static training dataset of the RL agent...The attacker can poison the training trajectories with a specified trigger and manipulate the victim agent’s reward". Such assumption is too strong since AD companies attach significance to the data for training. It is hard or even impossible to manipulate the whole dataset.
2. The poison ratio is too high. The experiments are done with a poison ratio of 15%. This is too high to be stealthy in terms of dataset preparation. Dataset checking is easy to find abnormal data.
3. The font in figures and tables are too small. For example, table 4 and figure 3 have too small font which make it hard to read the numerical results.

**Questions:**

1. Why not try one vehicle trajectory as triggers? This will make it easy to activate the backdoor.

---

### Official Review · Reviewer_ncHN · 2024-11-03

**Soundness:** 2
**Presentation:** 1
**Contribution:** 2
**Rating:** 3
**Confidence:** 3

**Summary:**

The paper proposes a new backdoor attack against end-to-end autonomous driving systems by trajectory-based dynamic multi-vehicle backdoor triggers. Their dynamic triggers are generated with temporal logic. This paper demonstrates that these dynamic triggers can deviate the trajectory of the victim vehicle based on the ADE and  MVR metrics. The paper also shows that their negative training strategy can improve the attack stealthiness and make them difficult to detect by existing defenses.

**Strengths:**

This paper has a novelty in the utilization of temporal logic to effectively explore the trajectory-based backdoor triggers. I can see that the evaluation metrics, ADE and MVR, are significantly compromised by their attacks. I also like the attitude of evaluating the robustness of their attacks on defense methods since just proposing attacks should not be ethical.

**Weaknesses:**

I have the following major concerns on this paper:

### The necessity of a dynamic multi-vehicle backdoor trigger is not well demonstrated

This paper should clarify that traditional backdoor attacks, such as static backdoor triggers, cannot be effective in this problem domain, end-to-end autonomous driving. Their idea of the dynamic multi-vehicle trigger is interesting to see, but the necessity needs to be well demonstrated as it requires much higher attack capability, i.e., a much stronger threat model. Just citing Gong et al. (2024) is not sufficient to support their claim. As they also mention Cao et al. (2022; 2023) and Zhang et al. (2022), these inference-time adversarial attacks could be enough to compromise the target. This paper should provide sufficient discussion and evaluation to demonstrate the advantages of their methods more clearly.

### The poisoning rate is too high considering the threat model

As this paper also admitted the limitation, the current poisoning rates (~15%) look too high to justify their attacks on end-to-end autonomous driving. For example, Suday drive in training data could be 15% (i.e., 85% is Mon-Sat driving). In this case, the current poisoning rates just look like they are building a "front door" rather than a backdoor. The acceptable poisoning rates could highly depend on the application domain. This paper should discuss or possibly demonstrate the feasible attack pipeline to a realistic end-to-end autonomous driving system.

### Lack of evaluation in physical-world and/or closed-loop environments

The current evaluation only with the ADE and MVR metrics is not sufficient to be fully convinced of the end-to-end attack consequences in practical driving scenarios. Furthermore, we still do not know if the generated triggers are actually realizable in the physical world and we do not know how their attacks are robust against real-world noises and the inaccuracies between design and realized trajectories. I understand that physical-world and closed-loop experiments are expensive and it is hard to conduct a large-scale experiment. However, this paper should provide at least a few representative successful case studies to prove the practicality and realizability of their attacks in real scenarios.

**Questions:**

- Are there any prior works saying that static or single-vehicle triggers are not effective in attacking RL agents in end-to-end autonomous Driving?
- Did you confirm the deployability and realizability of their attack in real-world driving scenarios?
- How did you choose the poisoning rates?

**Details Of Ethics Concerns:**

No concerns

---

### Official Review · Reviewer_Vgb5 · 2024-11-03

**Soundness:** 3
**Presentation:** 2
**Contribution:** 3
**Rating:** 5
**Confidence:** 3

**Summary:**

This paper proposes an innovative backdoor attack method for end-to-end autonomous driving systems, embedding backdoors in reinforcement learning (RL)-driven autonomous driving models through multi-vehicle trajectories as triggers. The authors utilize Temporal Logic (TL) to generate and evaluate trigger trajectories, and employ negative training to enhance the stealth and accuracy of the attack. Extensive experiments validate the effectiveness and flexibility of this approach in different environments, highlighting the limitations of existing defense mechanisms in countering such attacks. The contributions of this paper include introducing a new attack paradigm, developing a trajectory generation and evaluation framework, and conducting detailed experimental analysis.

**Strengths:**

1. The paper introduces a backdoor trigger mechanism based on multi-vehicle trajectories, breaking the limitations of traditional static triggers like images or markers and making detection by conventional defenses more challenging.
2. The trajectory generation framework combines behavior-driven generation with temporal logic evaluation, ensuring generated triggers meet strict temporal and spatial constraints efficiently and flexibly.
3. To improve attack stealth, the authors employ a "negative training" mechanism by adding non-trigger "patch trajectories," ensuring the model only activates the backdoor under true trigger conditions.

**Weaknesses:**

1. Lack of real-world validation as environmental factors like lighting and weather could affect sensor detection of trigger trajectories; further analysis on these effects would improve practical applicability.
2. Limited testing of defenses, as only data smoothing and DP-SGD were evaluated; more advanced defenses would provide a fuller assessment of this attack’s threat.
3. Restricted extensibility of the temporal logic framework, which uses a single specification; more varied temporal and spatial constraints could better support complex, stealthy attack patterns.

**Questions:**

1. How would the proposed attack perform under varying environmental conditions, such as heavy traffic, adverse weather, or low lighting? Could these factors impact the reliability of trigger detection?
2. How sensitive is the attack to minor variations in vehicle trajectories or timing? For example, would slight deviations in trigger trajectories still activate the backdoor reliably?

---

### Note · Authors · 2024-11-17

I have read and agree with the venue's withdrawal policy on behalf of myself and my co-authors.